# How Intuitive Is the Administration of Pediatric Emergency Medication Devices for Parents? Objective Observation and Subjective Self-Assessment

**DOI:** 10.3390/pharmacy12010036

**Published:** 2024-02-18

**Authors:** Ruth Melinda Müller, Birthe Herziger, Sarah Jeschke, Martina Patrizia Neininger, Thilo Bertsche, Astrid Bertsche

**Affiliations:** 1Department of Neuropaediatrics, Hospital for Children and Adolescents, University Medicine Rostock, Ernst-Heydemann-Strasse 8, 18057 Rostock, Germany; ruth_melinda.mueller.2@uni-leipzig.de (R.M.M.); birthe.herziger@med.uni-rostock.de (B.H.); sarah.jeschke@med.uni-greifswald.de (S.J.); astrid.bertsche@uni-greifswald.de (A.B.); 2Drug Safety Center, Leipzig University Hospital, Leipzig University, Brüderstrasse 32, 04103 Leipzig, Germany; martina.neininger@uni-leipzig.de; 3Department of Neuropaediatrics, Hospital for Children and Adolescents, University Medicine Greifswald, Ferdinand-Sauerbruch-Strasse 1, 17475 Greifswald, Germany; 4Clinical Pharmacy, Institute of Pharmacy, Medical Faculty, Leipzig University, Brüderstrasse 32, 04103 Leipzig, Germany

**Keywords:** pediatrics, intuition, dosage forms, medical devices, medication errors, emergencies, administration

## Abstract

Background: to assess the intuitiveness of parents’ administration of pediatric emergency devices (inhalation, rectal, buccal, nasal, and auto-injector). Methods: We invited parents without prior experience to administer the five devices to dummy dolls. We observed whether the parents chose the correct administration route and subsequently performed the correct administration procedures without clinically relevant errors. We interviewed parents for their self-assessment of their own administration performance and willingness to administer devices in actual emergencies. Results: The correct administration route was best for the inhalation device (81/84, 96% of parents) and worst for the intranasal device (25/126, 20%). The correct administration procedures were best for the buccal device (63/98, 64%) and worst for the auto-injector device (0/93, 0%). Their own administration performance was rated to be best by parents for the inhalation device (59/84, 70%) and worst for the auto-injector device (17/93, 18%). The self-assessment of the correct administration overestimated the correct administration procedures for all the devices except the buccal one. Most parents were willing to administer the inhalation device in an emergency (67/94, 79%), while the fewest were willing to administration procedures the auto-injector device (28/93, 30%). Conclusions: Intuitiveness concerning the correct administration route and the subsequent correct administration procedures have to be improved for all the devices examined. The parents mostly overestimated their performance. Willingness to use a device in an actual emergency depended on the device.

## 1. Introduction

Acute exacerbations of asthma, epileptic seizures, and anaphylaxis can have severe negative consequences for the health of children and adolescents and even lead to death [1,2,3]. With the administration of one single emergency medication, they can often be managed out of hospital up to stabilization [4,5,6]. However, the use of medications in emergencies takes place in extremely stressful situations. This is especially true for medical laypersons [7]. A special group in this regard are the parents of children to whom drugs are to be administered in a home environment [8]. Moreover, unlike nurses, parents have not usually undergone the necessary medical training. While specialist knowledge may not be required for first aid in an emergency situation, and while training parents might lead to an improvement in theoretical knowledge and practical skills, it is difficult to reproduce the exact conditions that occur in emergency situations in order to train parents in the administration of emergency medications. The careful reading of a package insert hardly seems practicable under emergency conditions. Additionally, the administration of medications by parents presents many challenges [9]. Special circumstances can increase these challenges even more, for example, in parents with intellectual disabilities [10]. It is therefore crucial that emergency medications are particularly intuitive to use by laypersons without any special theoretical knowledge or practical skill training. Intuition is defined as knowing or understanding without the conscious use of reasoning according to the MESH term’s definition [11]. Other definitions state that in the context of human–computer interactions “a technical system can be used intuitively in the context of a task to the extent that the respective user can interact effectively through the unconscious application of prior knowledge” [12]. The same authors [12] explain that prior knowledge and experience in particular can be used to operate a system correctly and satisfactorily without any explicit external help. Intuition is a combination of knowledge, ability, endurance, intellectual abilities, and discursive methods of recognition, as summarized by Zühlke in [13]. In the context of drug use, intuitiveness can be characterized by the ease with which a medication can be administered to achieve the intended effect and prevent any avoidable adverse drug reactions. In order to be intuitive, the administration process should be as feasible and successful as possible without any specific instructions, e.g., by being provided by a patient information leaflet or by teaching skills to parents, for example as part of a teaching session. Intuitiveness, however, is even more important in specific medications if, as with emergency devices, they are not to be administered by an experienced healthcare provider or by patients themselves but by accompanying medical laypersons. The use of emergency medications is further impeded by the fact that they are often complex devices. The first question is where exactly the device should be administered. In the second step, administration should also be correctly performed in order to be effective. Finally, and this should not be underestimated, it is also important that the complexity of the administration process does not deter a potential user from using it in an emergency. It should be borne in mind that an emergency situation creates additional psychological stress even if training has previously been provided. It should therefore be possible to use the medical device intuitively and without much thought or further explanation, ideally also without prior training, as even parents who have received training may not remember the contents of the training.

To explore intuitiveness with the administration of different medical devices used in emergencies, we assessed parents’ skills in administering placebo emergency medication devices to dummy dolls. Additionally, we interviewed parents on whether they thought they had used the device correctly and whether they were willing to administer the device in actual emergencies.

## 2. Materials and Methods

### 2.1. Setting and Study Design

After receiving ethic approval from the responsible local ethics committee (registration number A 2019-0069), this study was performed between December 2019 and April 2021. We invited parents to participate in our study during their child’s outpatient or inpatient stay in the pediatric department at a university hospital. Written informed consent was obtained from all participating parents. Sufficient German language skills to understand the content of the study were an inclusion requirement. Parents were enrolled for the following administration devices typically used for emergency medications and commercially available as follows: an inhalation device (metered-dose inhaler with spacer), a buccal device, a rectal device, an intranasal device, and an auto-injector device. The inclusion criterion for participation for each type of device was that parents were not allowed to have prior experience with the administration of the respective device for the administration of emergency medications.

Our study consisted of two methodologically distinct parts as follows:Objective observation by a monitoring: Participating parents were observed while demonstrating the use of the medical devices by administering them to dummy dolls. No time requirement or limit for administration existed. Observation was performed by the same monitor for all parents and all medical devices. The monitor was a pharmacist with several years of experience in pharmacy and advanced training in drug information. Drug administration to real patients was not performed in this study.Subjective self-assessment by a structured interview: we asked whether parents thought they had used the device correctly and whether they were willing to administer the device in a real emergency.

### 2.2. Objective Observation by Monitoring

The study addressed the administration of five different medical devices typically used for emergency medications by non-healthcare professionals in pediatrics as follows:Medical device for inhalation administration: a metered-dose inhaler with spacer as typically used for the treatment of acute dyspnea.Medical device for rectal administration: a rectal device as typically used for the treatment of prolonged acute convulsive seizures.Medical device for buccal administration: a buccal device as typically used for the treatment of prolonged acute convulsive seizures.Medical device for intranasal administration (syringe with mucosal atomization device): an intranasal device as typically used for the treatment of prolonged acute convulsive seizures.Medical device for auto-injector administration: a prefilled auto-injector device as typically used for the treatment of anaphylaxis.

The following aspects were considered for the monitoring.

For the purpose of our study, medical devices for emergency medications (placebos without active ingredients) were used. These corresponded to medical devices commercially available and approved for emergency medication. As the aim of our study was to investigate intuitiveness with the use of emergency medications, all medical devices were given to the participating parents in neutral packaging without a patient information leaflet or other drug information. Then, the parents had to determine how to use it on dummy dolls. This approach was chosen so that parents would have to use the medical devices intuitively without further information, prior training, or advice from a physician or pharmacist. Placebo devices were chosen to protect parents from exposure to active ingredients, particularly in the case of inappropriate administration. The order of medical devices presented to parents was randomized to minimize systematic errors.Quality assurance of the monitoring: An expert panel developed a checklist to document parental performance in administering the placebo devices. The expert panel consisted of a pediatrician/pediatric neurologist, four pharmacists with special expertise in clinical pharmacy and drug information, and a child and adolescent psychotherapist, all of whom were professionally involved in the use of emergency medications. The expert panel created a written checklist that was improved in several personal or video meetings and following e-mail correspondence. The checklist was piloted with seven volunteers before being used in the main study in which these volunteers did not participate.

The documentation of the administration processes based on the checklist was reviewed for intuitiveness as follows:We assessed the number of parents choosing the correct administration route (A.1).We assessed the number of parents performing correct administration procedures based on those choosing the correct route (A.2.a) and based on all enrolled parents (A.2.b).

Administration procedures were evaluated as correctly performed if no administration errors of moderate or high clinical relevance were observed according to an assessment of the expert panel. The panel used the Summary of Product Characteristics (i.e., the drug label) and additional material addressing the appropriate use (e.g., a patient information leaflet) of an originator product corresponding to the placebo administration device to predefine and classify the clinical relevance of the administration errors separately for each emergency medication device.

### 2.3. Subjective Self-Assessment by a Structured Interview

In a structured interview, all parents were asked for their self-assessment in the following items:Whether parents had used the device correctly (B).Whether parents were willing to use the device in a real emergency (C).

Development of the interviews: The interview questions were prepared in advance by the expert panel. The structured interview was then piloted by the same seven volunteers as in the development of the monitoring checklist. After piloting, the interview questions were adapted.

### 2.4. Statistical Analysis

For data analysis, Microsoft Excel (Microsoft Corporation, Redmond, WA, NY, USA) and SPSS (V29.0, IBM) were used. We performed a statistical test between error-free and error-prone processes depending on the professional group (healthcare professional or not) using the chi-square test or the Fisher’s exact test (depending on the appropriateness of the group size). Frequencies are reported as numbers and percentages, with continuous data as the median with the first (25%) and third (75%) quartile (Q25/Q75) and the range (minimum/maximum).

## 3. Results

### 3.1. Characteristics of Participating Parents

A total of 167 parents were invited to participate, and 126 (75%) of them gave their written, informed consent. From these 126, 84 (67%) had no experience with inhalation devices for emergency medications, 91 (72%) had no experience with rectal devices for emergency medications, and 98 (78%) had no experience with buccal devices for emergency medications. None of the 126 parents had any experience with intranasal devices for emergency medications, and 93/126 (74%) had no experience with auto-injector devices. These parents were enrolled for observation of the respective devices and for the structured interview on the respective medical devices. Their characteristics are shown in Table 1.

The statistical test between the error-free and error-prone processes depending on the two groups (healthcare professionals or not) achieved the following results: for inhalation, buccal, or rectal devices: n.s. (Chi-square test); for autoinjector devices: not assessable (since all participants made errors); for nasal devices: *p* = 0.001 (Fisher’s exact test).

### 3.2. Objective Observation

#### 3.2.1. Intuitiveness Assessed as Correctly Chosen Administration Route (A.1)

Intuitiveness in choosing the correct administration route was best for the inhalation device [81/84 (96%) parents indicating the correct administration route]. The buccal device was the second most intuitive [66/98 (67%)], followed by the rectal device [46/91 (50%)] and the auto-injector device [30/93 (32%)]. The intranasal device performed the worst in this category [25/126 (20%)]. The observed performance in choosing the correct administration route is shown in Table 2.

#### 3.2.2. Intuitiveness Assessed as Correctly Performed Administration after the Correct Administration Route Had Been Chosen Based on Those Choosing the Correct Route (A.2.a) and Based on All Enrolled Parents (A.2.b)

In this category, the buccal device performed best [63/66 (95%) correctly performed processes when the administration route was correctly chosen]. With the intranasal device, 21/25 (84%) administrations were performed correctly, followed by the rectal device [16/46 (35%)]. The inhalation device and the auto-injector device performed worst [8/81 (10%) and 0/30 (0%)]. Considering the correct administration route without any further clinical errors in the consecutive administration procedure, administration of the different medical devices was performed as follows: the inhalation device [8/84 (10%)], rectal device [16/91 (18%)], buccal device [63/98 (64%)], intranasal device 21/126 [(17%)], and auto-injector device [0/93 (0%)]. Detailed administration error rates with high and moderate clinical relevance as assessed by an expert panel are presented in Table 3.

### 3.3. Subjective Self-Assessment

#### 3.3.1. Self-Reported Performance (B)

Parents most frequently assessed their performance as correct for the inhalation device [59/84 (70%)] and least frequently for the auto-injector device [17/93 (18%)]. The complete results are shown in Table 2.

#### 3.3.2. Use Willingness in a Real Emergency (C)

Overall, 67/84 (79%) of respondents said they would actually use the inhalation device in a real emergency, which was the highest value for all the medical devices. The lowest value was reported for the auto-injector device [28/93 (30%)]. Details are shown in Table 2.

## 4. Discussion

### 4.1. General Considerations

We found that intuitive administration to a dummy doll by parents was improvable for all medical devices either regarding the correct administration route or the correct administration procedure. Most parents chose the correct administration route for the inhalation device and performed best regarding the correct administration procedure when administering the buccal device. Most parents overestimated their own performance. The willingness to administer a device depended on the device, but this was comparatively good in general.

### 4.2. Methodical Aspects

In our study, we deliberately refrained from enclosing a patient information leaflet or providing the opportunity to contact a healthcare professional for questions. Of course, these measures can help in real emergency situations and provide useful support for correct use. However, for the following reasons, we still consider it essential that users should be able to perform the administration as intuitively correct as possible without any accompanying information. First of all, it often happens that a patient information leaflet is simply not on hand in an emergency because the outer packaging, including the patient information leaflet, has been removed for transportation. What is more, users may not even understand the patient information leaflet due to limited language skills or a reading disability. This is exacerbated when intellectual competence is limited. In addition, the often rather legally formulated text overwhelms many “normal” users even when they are not in an emergency situation. In a stressful emergency, in the vast majority of cases a medical layperson is unlikely to be emotionally able to read the instructions in a patient information leaflet calmly. Furthermore, there is usually no physician available as a contact person at the onset of a medical emergency. All of this contributes to the fact that administration should be as intuitive as possible.

We did not analyze an anxiety or fear parameter in the context of this study. However, we assume that the level of stress in this experimental setup is likely to be significantly lower than that in a real emergency situation. We therefore see the parents’ potential anxiety as an important reason why administration should be as intuitive as possible. Only that which is intuitively carried out correctly can be applied in an emergency situation.

Another methodological aspect that should be considered when comparing different emergency devices is the complexity of their administration processes. It is obvious that an administration process consisting of several steps can lead to more errors than a process with comparatively few steps. For example, an inhaler requires more steps than a buccal device. This is also confirmed in our results. In the case of inhalation, the administration route was found to be correct in most observations. However, the subsequent administration process, which consists of numerous process steps, was not performed correctly very frequently. Obviously, complexity therefore limits intuitiveness with administration. In our study, the situation was different for the buccal device. The participants who chose the correct administration route usually also performed the further administration steps correctly, which were less complex than those for some other devices. From these considerations, it can be concluded that emergency medication should not only be intuitive but should also have as few administration process steps as possible. If not, many steps and complicated administration procedures often lead to incorrect use or need to be practiced and explained in detail in advance; then, they can be practiced regularly. This might work well for the regular use of an inhaler (but often this does not); however, in the case of emergency use by laypersons, a question arises as to the usefulness of such administration. If it cannot be used correctly, it cannot be effective in treating symptoms.

It was particularly important for us to ask the participants about their profession. It might be expected that working in the medical field led to better results in the correct use of the emergency devices. With regard to this aspect, the statistical tests did not reveal any significant difference between healthcare professionals and the other participants except for the nasal administration. Due to the low numbers, it is doubtful whether this difference can be considered valid in this sub-evaluation. Therefore, it is initially surprising that health professionals themselves do not perform better. This may also be due to the fact that we generally (also for healthcare professionals) excluded those who had direct experience with the emergency use of the respective device.

### 4.3. Inhalation Device

Analysis of the rate of the completely correctly performed administration routes and procedures showed that only 10% of the inhalation devices were ultimately used correctly. However, the overall self-assessment of correct use was 70%. Not only were administration errors very frequent but parents also overestimated their performance. As a consequence, it is not surprising that a high percentage of parents (79%) were willing to use an inhalation device in a real emergency despite a poor objective performance. This discrepancy between self-assessment and actual performance might be due to the fact that almost all parents chose the correct administration route. A previous study [14] highlights the need for education regarding the administration of inhalation devices, including practical training, to increase patient safety. Further studies claim that increasing patient compliance, minimizing the use of unnecessary excipients, and designing simple and self-intuitive inhalation devices that provide the user with good feedback on inhalation procedures are beneficial to patient safety [15]. The high percentage of parents choosing the correct administration route for inhalation devices suggests that they are in themselves intuitive devices—the many process steps, however, limit the intuitiveness considerably. Another publication [16] calls for the development of medical devices that reduce the complexity of the administration procedure and are intuitive and user-friendly. However, concrete concepts for the optimization of inhalation devices have only rarely been published until now, for instance in [17] where the authors measured the ease of use for healthcare professionals and users with and without experience with the inhaler.

The administration of inhalation devices is very complex and by no means intuitive because the many steps can be arranged in the wrong order, forgotten, or carried out incorrectly. Furthermore, these administration errors are often not recognized by users themselves. Consequently, skill deficits are frequently accompanied by knowledge deficits. It is unlikely that parents will actively seek out information, for instance, by asking the physician or pharmacist themselves. The focus rather ought to be upon actively providing information to parents.

### 4.4. Rectal Device

Upon an objective observation, the successful administration of the rectal device was similar to the performance observed for the inhalation device. Indeed, in 18% of the demonstrations, a correct administration procedure was observed, while the administration route was chosen correctly by half of the enrolled parents. Numerous errors, as recently reported in several studies [18,19,20], were recorded in our study. Nevertheless, the willingness to use the device in an actual emergency reported by 60% of parents barely exceeds the self-assessment of correct use. As with inhalation devices, parents should be actively coached in the administration of rectal devices. Errors in the administration procedure are often overlooked, especially when they are based on knowledge deficits that do not allow for conscious recognition of errors [21]. In our present study, we deliberately excluded previous experience with administration among the participating parents.

### 4.5. Buccal Device

The results for buccal administration are particularly interesting. Two thirds of parents intuitively chose the correct administration route. Of the participating parents, 95% performed the administration procedures correctly after having chosen the correct route. In 64% of all procedures, no clinically relevant errors were observed. In 51% of parents, the self-assessment of correct use lagged behind the objective performance. A total of 65% of parents recorded a willingness to use the device in an emergency. The error rate observed in previous studies [18,19,20], for example for anti-seizure rescue medication, was lower for the buccal devices than for the rectal devices. We can confirm this result in the study presented here. However, it is notable that in the current survey, self-assessment—with the exception of buccal devices—significantly overestimates competence in administration compared with the objectively observed performance. This is in line with previous studies [20] where 100% of the parents surveyed stated that they had never experienced any problems with the buccal administration route although numerous errors were observed. This underlines the importance of intuitive devices as incorrect use is frequently not identified by users themselves and therefore no support is sought from healthcare professionals. As a suggestion, pictograms could be used to clarify the intended use. In our opinion, this would be a reasonable—and probably sufficient—step to increase correct use and confidence in using the device.

### 4.6. Intranasal Device

The intranasal device scored poorly in the objective evaluation of the choice of the route of administration at 20% even though 84% of the subsequent administration procedures steps were performed correctly. However, only 17% of the overall administrations were correctly performed due to the difficulties in finding the correct administration route. Although intuition studies regarding nasal use are rare, a study [22] reported the following: to evaluate the ease of use, user preference, and effort required, nasal glucagon was compared with injectable glucagon in a simulation of severe hypoglycemia. In this randomized crossover study, dummy dolls were used in simulated high-stress environments. Trained and untrained participants performed the administration and then completed questionnaires. In terms of intuitiveness, it was particularly interesting that untrained users also found nasal administration to be more intuitive than injections that required reconstitution. However, at 29%, the self-assessment of correct use reflects the objective assessment fairly accurately and shows that parents can be expected to ask questions when problems arise. Nevertheless, the design of the devices should be improved, and additional guidance for administration should support the reasonably high willingness to use the device (39%). According to our study, intranasal administrations are among the least intuitive in terms of administration route. We recommend that pictograms be used on the packaging to make the administration route clear in a simple way.

### 4.7. Auto-Injector Device

Only 32 percent of the participants chose the correct administration route for the auto-injector device on the dummy, and no one performed the subsequent administration procedure correctly. A review [23] of epinephrine use highlighted knowledge gaps and a variety of handling errors regardless of the approach to epinephrine administration in the treatment of anaphylaxis. In our opinion, a contributing factor is the lack of intuition in the use of the auto-injector, which is often used the wrong way and thus leads to its administration into the user’s thumb; this can result in dangerous local and systemic effects for the user. This is also supported by a case study [24] in which the auto-injection was accidentally administered into the hand. Accidents in which administration was made to other parts of the body have also been reported [25] and highlight the problems with auto-injectors. A larger evaluation reported by a poison center confirmed those problems [26]. Moreover, the self-assessment of correct administration was 18%, revealing that many parents overestimated their performance. Of all participating parents, 30% were willing to administer the device in a real emergency. It is also difficult to explain why the willingness to use the device exceeds the rather modest self-assessment of the administration performance.

### 4.8. Comparison of the Devices Investigated

Even if it should be noted that not all devices are equally appropriate for all the active ingredients and emergency situations, in some cases there are options that should be weighed up against each other, such as the buccal and rectal administration of emergency anti-seizure medication [27].

When comparing the different devices, it is noticeable that the inhalation device is particularly impressive due to the high rate of intuitively found right administration routes compared with those of all the other devices. However, the use of this rather complex form of medication, which involves many different process steps, is associated with a much higher rate of administration errors than that of any other device examined. In self-assessment, on the other hand, inhalation is again ahead of all devices, which suggests a particularly high discrepancy between self-assessed and externally assessed competence compared with those observed for all other devices.

With less complex devices, i.e., those with fewer process steps, which are accordingly also easier to use, it can be seen that administration is carried out without major problems once the location is correct, as can be observed for the buccal device. The comparatively complex auto-injector performs worst out of all the devices tested here in terms of both practical and self-estimated performance.

A comparison of the buccal and rectal devices is particularly interesting. Although the self-assessment is comparable for both, the rectal device scored numerically worse in all the categories of the external evaluation. This indicates that the buccal device can be preferred choice, e.g., in seizure events.

Apart from this example for anti-seizure medications, pharmaceutical devices are often not interchangeable in terms of the active ingredients and indications. If we try to find the best devices with regard to the aspect of intuitiveness, it is therefore difficult to make a simple statement. There are devices where the right place is simply found, but the subsequent administration is complex and error-prone and therefore not intuitive. With others, such as the buccal device, it is the other way round. Once it is administered in the right place, the administration itself is rather simple. Therefore, it will not be easy to develop one intuitive drug form.

### 4.9. Strategies for Improving Intuitiveness: A Comparison with the Literature

McCaughey et al. [28] performed an online survey among 6298 school nurses about the availability, safety, and use of epinephrine auto-injectors, albuterol inhalers, and glucagon for emergencies in particular. They reported that school nurses are a guarantee for the safety of students at school and should be further promoted. Such concepts can ensure that trained personnel are available on site. However, in our opinion, this should not replace the goal of developing intuitive devices that can be used without prior teaching and training. Finally, the circle of persons potentially administering emergency medications should be expanded as specialized professions are not present in all life situations. In order to expand the user group of the system, other people, such as unlicensed, assistive personnel [29], have already been included in the handling of a glucagon emergency device. This underlines the advantages of intuitive usability for users in addition to the specialist medical professions.

The desirable intuitiveness regarding the use of medical devices is not restricted to children in their school environment but also affects adults. For example, buccal midazolam solutions were used off-label in adults [30]. Difficulties in administration were reported by 13% of the adult patients, indicating that administration problems caused by unintuitive use might play a key role in safe drug use in adults too.

As part of the transition of our pediatric patients into adulthood, we are planning to carry out an investigation dealing with emergency medicines in adults with a focus on specially abled adults.

An important question arises about possible measures to improve the safety of drug administration in addition to improving intuitiveness with the use of medical devices in the future. Apps could be one possibility, such as one developed for an out-of-hospital cardiac arrest cardiopulmonary resuscitation scenario for the use of different drugs by paramedics with drug preparation autonomy [31]. Compared with traditional methods, the use of this app significantly reduced the rate of medication errors and the time to medication delivery when preparing emergency medications in a prehospital setting [31]. However, this requires the respective app to be downloaded and available in an emergency. Whether this will actually be the case under real circumstances can at least be questioned underlining the demand for the intuitive usability of emergency devices.

A study by Shultz et al. [32] is interesting with regard to the intuitiveness with not only the devices themselves but also the abbreviations used in their preparation. Here, for example, “ER” was interpreted as “emergency release” rather than “extended release” and was thus misclassified as a short-acting drug. This shows the limits of intuition. Misinterpretations can lead to misinterpretations of seemingly obvious items. This should also be considered in the development of medical devices.

### 4.10. Limitations

This study has limitations that should be considered when interpreting the results as follows: Firstly, the study was monocentric. Secondly, our study was intentionally not conducted in a real emergency situation. Thirdly, almost one third of the participating parents were employed in health-related or educational professions. Fourthly, in the case of the inhalation device, we counted activation as a successful process step to allow for comparison with other administration devices. Moreover, actual inhalation could not be studied using the dummy doll.

## 5. Conclusions

To our knowledge, this study constitutes the first exploration of the intuitive use of administration devices for pediatric emergency medication by an observation accompanied by self-assessment through a structured interview. We found that intuitiveness with commercially available administration devices for emergency medications differed considerably. Intuitiveness was limited for all devices, which was most especially so for the auto-injector device. This was accompanied by a poor self-assessed performance and limited willingness to actually administrate it in an emergency situation.

## Figures and Tables

**Table 1 pharmacy-12-00036-t001:** Characteristics of parents participating in the monitoring and in the interview.

	Inhalation Device	Rectal Device	Buccal Device	Intranasal Device	Auto-Injector Device
**Total number [*n*]**	84	91	98	126	93
**Age [years]**					
Median	36.5	36	35.5	36	36
Q25/Q75	33/41	32/41	32/40	33/41	32/40
**Gender [*n* (%)]**					
Male	18 (21%)	17 (19%)	17 (17%)	22 (17%)	16 (17%)
Female	66 (79%)	74 (81%)	81 (83%)	104 (83%)	77 (83%)
**Profession [*n* (%)]**					
Healthcare	11 (13%)	11 (12%)	18 (18%)	25 (20%)	15 (16%)
Education	8 (10%)	10 (11%)	11 (11%)	15 (12%)	13 (14%)
Others	65 (77%)	70 (77%)	69 (70%)	86 (68%)	65 (70%)

**Table 2 pharmacy-12-00036-t002:** **Observation:** Number of parents choosing the **correct administration route** (A.1). Number of parents performing **correct administration procedures** based on those choosing the correct route (A.2.a) and based on all enrolled parents (A.2.b). We observed parents who had no previous experience with the particular medical device but chose the correct administration route intuitively. **Interview:** Number of parents **self-assessing** their administration route **and procedure** as **correct** (B). Number of parents self-assessing that they were willing to actually use it in real emergencies (C). We interviewed parents participating in the observation.

	Observation (A)			Interview (B, C)	
Number of Parents…	…Choosing the Correct Administration-Route (A.1) Based on All Observed Procedures	…Performing Correct Administration Procedures Based on Those Choosing the Correct Route (A.2.a)	Based on All Enrolled Parents (A.2.b)	…Self-Assessing Their Administration Route and Procedure as Correct (B)	…Self-Assessing that They Were Willing to Actually Use in Real Emergencies (C)
**Inhalation device**	81/84 (96%)	8/81 (10%)	8/84 (10%)	59/84 (70%)	67/84 (79%)
**Rectal** **device**	46/91 (50%)	16/46 (35%)	16/91 (18%)	48/91 (53%)	55/91 (60%)
**Buccal** **device**	66/98 (67%)	63/66 (95%)	63/98 (64%)	50/98 (51%)	64/98 (65%)
**Intranasal device**	25/126 (20%)	21/25 (84%)	21/126 (17%)	36/126 (29%)	49/126 (39%)
**Auto-injector device**	30/93 (32%)	0/30 (0%)	0/93 (0%)	17/93 (18%)	28/93 (30%)

**Table 3 pharmacy-12-00036-t003:** Number of parents committing administration errors of high and moderate clinical importance assessed by an expert panel (after having chosen the correct administration route) while administering several devices (without active ingredients) to dummy dolls without prior guidance or experience with the respective device.

Administration Device	Administration Error as Defined by an Expert Panel	Number of Parents Committing the Defined Administration Error after Choosing the Correct Administration Route (A.1, See Line 2 of Table 2)
Inhalation device	Protective cap not removed	7/81 (9%)
	Spacer not used	18/81 (22%)
	No sealed fit of mask/mouthpiece	23/81 (28%)
	Suspension MDI not shaken before actuation	69/81 (85%)
	Inhaler not operated (frequently enough)	13/81 (16%)
Rectal device	Protective cap not removed	2/46 (4%)
	Applicator not squeezed	7/46 (15%)
	Not removed under compression	30/46 (65%)
Buccal device	Protective cap not removed	2/66 (3%)
	Applicator not activated (pressed down)	2/66 (3%)
Intranasal device	Protective cap not removed	1/25 (4%)
	Mucosal atomization device not placed on syringe	0/25 (0%)
	Applicator not activated (pressed down)	3/25 (12%)
Auto-injector device	Protective cap not removed	10/30 (33%)
	Applicator not released	18/30 (60%)
	Applicator removed immediately after activation	28/30 (93%)
	Injection in finger of user	13/30 (43%)

## Data Availability

The data can be requested from the corresponding author upon reasonable request.

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
