# Peer review of "How Intuitive Is the Administration of Pediatric Emergency Medication Devices for Parents? Objective Observation and Subjective Self-Assessment"

_pharmacy, 2024, doi:10.3390/pharmacy12010036_

Round 1

Reviewer 1 Report

Comments and Suggestions for Authors

- Is it a very innovative study. It could be extended to caregivers of specially abled adult individuals as well.

- I suggest removing all parents who were employed in health care. They could be skewing the results.

- You could perform Kruskal Wallis on continous or Fisher's on categorical in Table 1. That will definitely strengthen the paper.

- Table 3: Very helpful table

-Were all the parents fluent in German and the documentation accompanying the medication available in German? If you could provide that information it could indicate if language is one of the factors that affects parent's understanding of how a potentially life saving drug can be used. 

Author Response

Pharmacy Editorial Office
MDPI, St. Alban-Anlage 66, 4052 Basel, Switzerland
Tel.: +41 61 683 77 34 (MDPI HQ Basel, Switzerland / 9:00 - 17:00 CET)
[email protected]
Managing Editor

Mr. Gary Zhang

Point-by-point answers to the Editor‘s and Reviewers‘ comments

Dear Dr. Zhang,

We would like to thank you and the reviewers very much for your work. Your constructive comments have enabled us to revise our manuscript in a targeted manner. We have also reconsidered a few points and discussed them again in the author team. In this way, we were able to sharpen the points where our wording wase not yet precise enough to be properly understood. This was very helpful for us.

We have answered the comments below point by point and have indicated which passages in the manuscript have been rephrased.

With kind regards,

On behalf of for all authors

Thilo Bertsche

Editor

  1. We kindly request that you consider revising your manuscript to include a
    minimum of 30 references to provide our readers with a comprehensive and
    well-supported body of knowledge.  Increasing the number of references will
    strengthen the scholarly foundation of your work and enhance its value to the
    readership.

=> We have expanded the references to 32 sources.

  1. We suggest the original research article or comprehensive review with a
    suggested minimum word count of 4000 words.

=> According to your advice, we have expanded our main text to over 4,000 words and in this way, in accordance with the reviewers’suggestions, we can present and discuss some aspects in more detail.

In order to comply with the Editor's comment, we have added some text passages at the request of the Reviewers and thus deepened the explanations of our results. These passages are explained in detail in the corresponding sections were we have addressed the Reviewers' comments. In addition, we have supplemented some passages and discussion points as follows in the manusript:

… in the introduction:

Intuition is defined as knowing or understanding without conscious use of reasoning according to the MESH-term definition [11]. Other definitions state in the context of human-computer interaction that "a technical system can be used intuitively in the context of a task to the extent that the respective user can interact effectively through the unconscious application of prior knowledge" [12]. The same authors [12] explain that prior knowledge and experience in particular can be used to operate a system correctly and satisfactorily without explicit external help. Intuition is a combination of knowledge, ability, endurance, intellectual abilities, and discoursive methods of recognition as summarized by Zühlke in [13]. In the context of drug use, intuitiveness can be characterized by the ease with which a medication can be administered to achieve the intended effect and prevent avoidable adverse drug reactions. In order to be intuitive, the administration process should be as feasible and successful as possible without specific instructions, e.g. provided by a patient information leaflet, or trained skills, e.g. from a teaching session. Intuitiveness, however, is even more important in specific medications if, as with emergency devices, they are not to be administered by an experienced healthcare provider or by patients themselves but by accompanying medical laypersons. The use of emergency medications is further impeded by the fact that they are often complex devices. The first question is where exactly the device should be administered. In the second step, the administration should also be correctly performed in order to be effective. Finally, and this should not be underestimated, it is also important that the complexity of the administration process does not deter a potential user from using it in an emergency. It should be borne in mind that an emergency situation creates additional psychological stress, even if training has previously been provided. It should therefore be possible to use the medical device intuitively and without much thought or further explanation, ideally alsowithout prior training, as even parents who have received training may not remember the content of the training.

… in the discussion a paragraph about “Methodical aspects”:

In our study, we deliberately refrained from enclosing a patient information leaflet or providing the opportunity to contact a healthcare professional for questions. Of course, these measures can help in real emergency situations and provide useful support for a correct use. However, for the following reasons, we still consider it essential that users should be able to perfom the administration as intuitively correct as possible without any accompanying information: First of all, it often happens that a patient information leaflet is simply not on hand in an emergency because the outer packaging, including the patient information leaflet, has been removed for transportation. What is more, users may not even understand the patient information leaflet due to limited language skills or a reading disability. This is exacerbated when intellectual competence is limited. In addition, the often rather legally formulated text overwhelms many “normal” users even when they are not in an emergency situation. In a stressful emergency, in the vast majority of cases a medical layperson is unlikely to be emotionally able to read the instructions in a patient information leaflet calmly. Furthermore, there will usually be no physician available as a contact person at the onset of a medical emergency. All of this contributes to the fact that the administration should be as intuitive as possible.

We did not analyze an anxiety or fear parameter in the context of this study. However, we assume that the level of stress in this experimental setup is likely to be significantly lower than in a real emergency situation. We therefore see the parents' potential anxiety as an important reason why an administration should be as intuitive as possible. Only what is intuitively done correctly can be applied in an emergency situation.

Another methodological aspect that should be considered when comparing different emergency devices is the complexity of their administration processes. It is obvious that an administration process consisting of several steps can lead to more errors than a process with comparatively few steps. For example, an inhaler requires more steps than a buccal device. This is also confirmed in our results: In the case of inhalation, the administration-route was found correctly in most observations. The subsequent administration-process, however, which consists of numerous process steps was very frequently not performed correctly. Obviously, complexity therefore limits the intuitiveness of the administration. In our study, the situation was different for the buccal device. Those participants who chose the correct administration-route usually also performed the further administration steps, which were less complex than for some other devices, correctly. From these considerations, it can be concluded that emergency medication should not only be intuitive, but also have as few administration process steps as possible. If not, many steps and complicated administration procedures often lead to incorrect use or need to be practiced and explained in detail in advance - and then practiced regularly. This might work well for the regular use of an inhaler (but often it does not) - however, in the case of emergency use by laypersons, the question arises as to the usefulness of such an adminsitration. If it cannot be used correctly, it cannot be effective in treating symptoms.

It was particularly important for us to ask the participants about their profession. It might be expected that working in the medical field led to better results in the correct use of the emergency devices. With regard to this aspect, the statistical tests did not reveal any significant difference between healthcare professionals and the other participantsexcept for the nasal administration. Due to the low numbers, it is doubtful whether this difference can be considered valid in this sub-evaluation. Therefore, it is initially surprising that health professionals themselves do not do better. This may also be due to the fact that we generally (also for health care professionals) excluded those who had direct experience of emergency use of the respective device.”

… in the discussion a paragraph about “Comparison of the devices investigated”:

Even if it should be noted that not all devices are equally appropriate for all active ingredients and emergency situations, in some cases there are options that should be weighed up against each other, such as buccal and rectal administration of emergency anti-seizure medication [27].

When comparing the different devices, it is noticeable that the inhalation device is particularly impressive due to the high rate of intuitively found right administration-routes compared to all other devices. However, the use of this rather complex form of medication, which involves many different process steps, is associated with a much higher rate of administration errors than any other device examined. In self-assessment, on the other hand, inhalation is again ahead of all devices, which suggests a particularly large discrepancy between self-assessed and externally assessed competence compared to all other devices.

With less complex devices, i.e. those with fewer process steps, which are then also easier to carry out, it can be seen, as with the buccal device, that once the location is correct, the administration is carried out without major problems. The comparatively complex auto-injector performs worst of all the devices tested here in terms of both practical and self-estimated performance.

A comparison of the buccal and rectal devices is particularly interesting. Althoughthe self-assessment is comparable for both the rectal device scorednumerically worse in all categories of the external evaluation.. This indicates that the buccal form can be preferred, e.g. in seizure events.

Apart from this example in anti-seizure medications, pharmaceutical devices are often not interchangeable in terms of the active ingredients and indications. If we try to find the best devices with regard to the aspect of intuitiveness, it is therefore difficult to make a simple statement. There are those devices where the right place is simply found, but the subsequent administration is complex and error-prone and therefore not intuitive. With others, such as the buccal device, it is the other way round. Once it is administered in the right place, the administration itself is rather simple. Therefore, it will not be easy to develop one intuitive drug form.

… in the discussion a paragraph about “Strategies for improving intuitiveness: a comparison to the literature”:

McCaughey et al. [28] performed an online survey among 6,298 school nurses about availability, safety and use of epinephrine auto-injectors, albuterol inhalers and glucagon for emergencies in particular. They reported that school nurses are a guarantee for the safety of students at school and should be further promoted. Such concepts can ensure that trained personnel are available on site. However, in our opinion, this should not replace the goal of developing intuitive devices that can be used without prior teaching and training. Finally, the circle of persons potentially administering emergency medicationsshould be expanded, as specialized professions are not present in all life situations. In order to expand the user group of the system, other people, such as unlicensed assistive personnel [29], have already been included in the handling of a glucagon emergency device. This underlines the advantages of intuitive usability for users beside the specialist medical professions.

The desireable intuitiveness regarding the use of medical devices is not restricted to children in their school environment, but also affects adults. For example, buccal midazolam solutions were used off-label in adults [30]. Difficulties in administration were reported by 13% of the adult patients indicating that also in adults administration problems caused by an unintuitive use might play a key role in safe drug use.

As part of the transition of our pediatric patients into adulthood, we are planning to carry out an investigation dealing with emergency medicines in adults with a focus on specially abled adults

An important question arises about possible measures to improve administration drug safety, in addition to improving the intuitiveness of the use of medical devices in the future. One possibilty could be apps, such as developed for an out-of-hospital cardiac arrest cardiopulmonary resuscitation scenario for the use of different drugs by paramedics with drug preparation autonomy [31]. Compared to traditional methods, the use of this app significantly reduced the rate of medication errors and the time to medication delivery when preparing emergency medications in the prehospital setting [31]. However, this requires that the respective app is downloaded and available in an emergency. Whether this will actually be the case under real circumstances can at least be questioned underlining the demand for intuitive usablity of emergengency devices.

A study by Shultz et al. [32] is interesting with regard to the intuitiveness of not only the device themselves, but also of the abbreviations used in their preparation. Here, for example, “ER” was interpreted as “emergency release” rather than “extended release”, and thus misclassified as a short-acting drug. This shows the limits of intuition. Misinterpretations can lead to misinterpretations of seemingly obvious items. This should also be considered in the development of medical devices.

  1. During our standard checks of all submissions, we noticed that the
    author's self-citation exceeds the optimal limit, which should not be more
    than 15%. Following COPE guidelines
    (
    https://publicationethics.org/node/44351), authors should not engage in
    excessive self-citation of their work. However, we would like to note that
    the receipt of this email does not imply that the self-citations in the
    article are invalid. Self-citations may be valid if, for example, they are
    needed to understand the background and history of the work in question. If
    you feel that the citations to the previous works are essential, an Academic
    Editor will check the appropriateness of these citations.

=> We have checked the references and by expanding further sources we now have a share of 5 out of 32 references (15.62%).

  1. This manuscript contains case studies using patients or survey
    respondents. The approval for publication from the patients or survey
    respondents is mandatory. Please provide a blank form of consent for
    publication. You can send a scan in PDF or Word format, whichever suits you
    better. This is only for our records and will not be made publicly available.

=> We attached the written informed consent form in the German original version (see “Attachments” to this letter).

Reviewer #1

1.1. Is it a very innovative study. It could be extended to caregivers of specially abled adult individuals as well.

=> We thank Reviewer #1 very much for this good advice and could imagine focusing the project on this specific group of caregivers of adult patients in the next step. The previous experience from this study, which was conducted in pediatric and adolescent medicine, would indeed be very helpful for this continuation in another group. To consider this remark we added a sentence as follows: “As part of the transition of our pediatric patients into adulthood, we are planning to carry out an investigation dealing with emergency medicines in adults with a focus on specially abled adults.”

1.2. I suggest removing all parents who were employed in health care. They could be skewing the results.

=> We would also like to thank the reviewer for this important remark. We excluded all individuals with experience with the respectitive device irrespective if the experience was a professional or a personal one. By excluding all health-care professionals, we would have excluded people relatively arbitrarily without being linked to concrete experiences with the respective device. We wanted to avoid this. Thus, we only excluded those in the analysis of the respective devices who were already familiar with the specific administration.  In order to take this Reviewer's comment into account, we carried out a statistical test comparing the health care group to the other participants.

To do so, we calculated a Chi square/Fisher's Excact test and compared error-free processes against error-prone processes in all device groups with regard to the influence of whether someone worked asa health care professional or not. For inhalation, buccal and rectal devices we found no difference between the groups in the Chi square test. For the autoinjector, an evaluation was not possible since errors were committed by all participants. There is a difference for nasal use, p=0.001 in the Fisher test. We added this information in the manuscript as follows: “The statistical test between error-free and error-prone processes depending on the two groups (healthcare professionals or not) achieved the following results: For inhalation, buccal or rectal devices: n.s. (Chi-square test). For autoinjector devices: not assessable (since all participants made errors; for nasal devices: p=0.001 (Fisher’s exact test).” And in the methods section: “We performed a statistical test between error-free and error-prone processes depending on the professional group (healthcare professional or not) using the chi-square test or the Fisher’s exact test (depending on the appropriateness of the group size).”

1.3. You could perform Kruskal Wallis on continous or Fisher's on categorical in Table 1. That will definitely strengthen the paper.

=>  Since all groups are subgroups from a total sample, the data are partly paired (the five groups are not independent but from the same sample of parents only selected by their prior experience in certain devices with multiple categories possible), which is why we refrained from a statistical analysis. Nevertheless, we have carried out Wilcoxon and Chi square  tests for experimental reasons and found no significance. For this reason, we would like to refrain from presenting those in the revision.

1.4. Table 3: Very helpful table

=> Thank you for the comment.

1.5. Were all the parents fluent in German and the documentation accompanying the medication available in German? If you could provide that information it could indicate if language is one of the factors that affects parent's understanding of how a potentially life saving drug can be used. 

=> Patients were enrolled as they were able to understand the verbal and written study information in German and gave their verbal and written informed consent. We supplemented the corresponding explanations as follows in the methods section: "Sufficient German language skills to understand the content of the study were an inclusion requirement.” Besides, we did not use any linguistic or other drug information materials in our study; we have made this clear in the following sentence in the method objective: “As the aim of our study was to investigate intuitiveness of the use of emergency medications all medical devices were given to the participating parents in neutral packaging without a patient information leaflet or other drug information.”

Reviewer #2

2.1. Study design (lines 99-103):  If I understand this correctly, subjects were not given the standard packages for these drugs/devices, and were not given any indication of how to use them (no diagrams, no written instructions). Instead, it seems that subjects were asked to infer proper use solely from an examination of the device. But why? What we want to know is whether parents would use these devices properly in the real world, a world in which packages would typically have diagrams and instructions. But this study is examining something very different: how parents would behave in a hypothetical world in which drugs/devices are packaged without instructions for use (or, at least, a world in which devices are always separated from their packaging). This makes the study less valuable. Why not combine this with a study of how well (and perhaps how quickly) parents can use the devices when standard packaging is present? (Lines 228-230 raise this issue, noting that parents are unlikely to seek guidance from healthcare professionals. True, but surely they'll look at the device packaging -- packaging that was not present for them in this study. The authors themselves make a similar point in lines 258-260.) This should be noted in the Limitations section of the manuscript.

=> We thank the reviewer for this critical assessment. Based on our many years of experience, however, we believe that administration of emergency devices should be intuitively possible without instructions. We have included the following arguments in the discussion section and have now better explained why we performed our study without using the patient information leaflet: “In our study, we deliberately refrained from enclosing a patient information leaflet or providing the opportunity to contact a healthcare professional for questions. Of course, these measures can help in real emergency situations and provide useful support for a correct use. However, for the following reasons, we still consider it essential that users should be able to perfom the administration as intuitively correct as possible without any accompanying information: First of all, it often happens that a patient information leaflet is simply not on hand in an emergency because the outer packaging, including the patient information leaflet, has been removed for transportation. What is more, users may not even understand the patient information leaflet due to limited language skills or a reading disability. This is exacerbated when intellectual competence is limited. In addition, the often rather legally formulated text overwhelms many “normal” users even when they are not in an emergency situation. In a stressful emergency, in the vast majority of cases a medical layperson is unlikely to be emotionally able to read the instructions in a patient information leaflet calmly. Furthermore, there will usually be no physician available as a contact person at the onset of a medical emergency. All of this contributes to the fact that the administration should be as intuitive as possible.”

2.2. Table 3:  The last column (number of parents making errors) is confusing. Where do the denominators of these fractions come from? Why are they not the same as the n for each device as shown in table 1? More explanation would help here.

=> Thank you for the comment. We evaluated only those administration processes for errors for which the correct application location had already been selected. These figures are shown in Table 2, and we now also explicitly point this out in Table 3. To consider this comment, we added the following item in Table 3: “Number of parents committing the defined administration error after choosing the correct administration-route (A.1, see line 2 of Table 2) ”

2.3. This paper makes a new contribution to the literature on proper use of drug administration devices, but because study participants were not provided with the device packaging the significance of this contribution is somewhat diminished. This study answers the question, "If you were given this device and told nothing about it, could you use it correctly?" But was anyone asking that question? Aren't we more interested in this question: "When given a device in its usual packaging, could you use it correctly?"

=> Thank you for the comment. As this deals with the same aspect as in comment 2.1, we refer here to the detailed answer and the comprehensive modification of the manuscript in this respect in point 2.1.

2.4. Although this contribution is modest, it's still useful. It is worth publishing, but it's a close call.

=> We hope that we now meetthe Reviewer's expectations, especially with our detailed response in 2.1. and the extensive revision of the manuscript in this respect, and would like to thank Reviewer #2 again for the critical comments.

Reviewer #3

The manuscript performed a detailed study on objective observation and subjective self-assessment of pediatric emergency 2 medication devices for parents. The perspectives of the study are thorough and considerate. Only a few things need to be addressed:

3.1 Did the study consider parental anxiety level as a common reason for not conducting the procedures?

=> We thank the reviewer very much for this important aspect and added to the discussion: “We did not analyze an anxiety or fear parameter in the context of this study. However, we assume that the level of stress in this experimental setup is likely to be significantly lower than in a real emergency situation. We therefore see the parents' potential anxiety as an important reason why an administration should be as intuitive as possible. Only what is intuitively done correctly can be applied in an emergency situation.”

3.2. In General Considerations, can authors conclude that as the complexity of the procedure or devices increases, the rate of parental intuition decreases?

=> We also agree with the reviewer that this is particularly interesting. Although we did not measure the complexity numerically, differences in complexity between, for example, a buccal device and an inhalation device are obvious. We have now taken this into account more comprehensively in the manuscript as follows:

“Another methodological aspect that should be considered when comparing different emergency devices is the complexity of their administration processes. It is obvious that an administration process consisting of several steps can lead to more errors than a process with comparatively few steps. For example, an inhaler requires more steps than a buccal device. This is also confirmed in our results: In the case of inhalation, the administration-route was found correctly in most observations. The subsequent administration-process, however, which consists of numerous process steps was very frequently not performed correctly. Obviously, complexity therefore limits the intuitiveness of the administration. In our study, the situation was different for the buccal device. Those participants who chose the correct administration-route usually also performed the further administration steps, which were less complex than for some other devices, correctly. From these considerations, it can be concluded that emergency medication should not only be intuitive, but also have as few administration process steps as possible. If not, many steps and complicated administration procedures often lead to incorrect use or need to be practiced and explained in detail in advance - and then practiced regularly. This might work well for the regular use of an inhaler (but often it does not) - however, in the case of emergency use by laypersons, the question arises as to the usefulness of such an adminsitration. If it cannot be used correctly, it cannot be effective in treating symptoms. It was particularly important for us to ask the participants about their profession. It might be expected that working in the medical field led to better results in the correct use of the emergency devices. With regard to this aspect, the statistical tests did not reveal any significant difference between healthcare professionals and the other participantsexcept for the nasal administration. Due to the low numbers, it is doubtful whether this difference can be considered valid in this sub-evaluation. Therefore, it is initially surprising that health professionals themselves do not do better. This may also be due to the fact that we generally (also for health care professionals) excluded those who had direct experience of emergency use of the respective device.”

3.3. Can authors make a conclusive discussion why the different devices draw different reactions from the parents?

=> In connection with the point mentioned in 3.2, we have now compared the various devices in a further chapter and have also taken better account of aspects of complexity.

3.4. Comments on the Quality of English Language

Minor editing of English language required.

=> The manuscript had been language edited by a native speaker. During the revision we aimed at clarifying some points by improving the wording.

Attachments

Teilnehmenden-Information

Teilnehmende-Information an dem Forschungsvorhaben

Untersuchung der intuitiven Anwendung von Notfallarzneimitteln durch medizinische Laien

 Sehr geehrte Damen und Herren,

vielen Dank, dass Sie sich für dieses Forschungsvorhaben interessieren!

Die Teilnahme an dem Projekt ist freiwillig und kostenfrei. Eine Vergütung für die Teilnahme wird nicht gewährt. Die Teilnahme hat KEINEN Einfluss auf die Behandlung Ihres Kindes in der Universitätskinder- und Jugendklinik Rostock oder Ihr Studium an der Universität Rostock. Sie können Ihre Einwilligung jederzeit, ohne Angabe von Gründen und ohne Nachteil für Sie oder Ihre Familie zurückziehen. Daten, die bereits anonym in die wissenschaftliche Auswertung eingeflossen sind, können gegebenenfalls nachträglich nicht mehr gelöscht werden. Sie werden in dieses Vorhaben nur dann einbezogen, wenn Sie dazu schriftlich Ihre Einwilligung erklären.

  1. Warum wird dieses Projekt durchgeführt?

Ärztlich verschriebene Notfallmedikamente können durch medizinische Laien in Notfallsituationen angewendet werden. Auf diese Weise kann im Einzelfall Leben gerettet werden. Dies gelingt jedoch nur dann, wenn die Anwendung richtig erfolgt. Daher sollten solche Notfallmedikamente insbesondere auch intuitiv anwendbar sein. Das heißt sie sollen ohne große Erklärung richtig angewendet werden können – selbst in einer Stresssituation wie einem Notfall beim eigenen Kind. Zur Verbesserung der Arzneimittelsicherheit von Notfallarzneimitteln zur Laienanwendung sollen mit diesem Projekt die Darreichungsformen der Notfallarzneimittel bei Anwendung im Hinblick auf diese intuitive Anwendbarkeit verglichen werden. Die gewonnenen Daten werden im Rahmen einer wissenschaftlichen Veröffentlichung und einer Promotionsarbeit anonym ausgewertet.

  1. Wie läuft das Projekt ab?

Nach Aufklärung und schriftlicher Einwilligung werden Ihnen nacheinander verschiedene Arzneiformen gereicht, die Sie in einer simulierten Notfallsituation an einer Demonstrationspuppe anwenden sollen. Sie werden dabei von der Durchführenden beobachtet. Nach jeder Arzneiform werden Sie befragt, in welchem medizinischen Notfall das entsprechende Arzneimittel nötig sein könnte. Im Anschluss an die Beobachtungsphase folgt die Befragung zu sozioökonomischem und medizinischem Hintergrund, Erfahrung mit Arzneimitteln und Präferenz für Darreichungsform von Notfallmedikamenten. Die Durchführungsdauer beträgt durchschnittlich 15 Minuten.

  1. Welchen Nutzen und welches Risiko geht mit dem Projekt einher?

Im Rahmen der Durchführung stehen Informationen zu den jeweiligen Notfallmedikamenten und Notfallsituationen zur Verfügung. Es besteht nach der Befragung die Möglichkeit zum Gespräch mit Arzt oder Apotheker.

Da es sich um eine Beobachtungs- und Befragungsstudie handelt, sind keine Risiken für die Probanden zu erwarten. Die zu verwendenden Darreichungsformen sind nadelfrei und enthalten Placebo, also keinen Wirkstoff.

  1. Welche Daten sind notwendig und was passiert mit den Daten?

Ihre persönlichen Daten werden im Rahmen des Interviews von Mitarbeitern der Universitätsmedizin erhoben, gesammelt und an einem gesicherten Ort des Klinikums gespeichert. Die Erfassung und Nutzung der Daten unterliegt den geltenden Datenschutzrichtlinien (EU-Datenschutzgrundverordnung); die Rechtsgrundlage der Datenverarbeitung ist Ihre Einwilligung (Art. 6 Abs. 1 Buchstabe a DSGVO).

Die erhobenen Daten werden anonymisiert, d. h. personenbezogene Daten werden derart verändert, dass die Einzelangaben über persönliche oder sachliche Verhältnisse nicht mehr oder nur mit einem unverhältnismäßig großen Aufwand an Zeit, Kosten und Arbeitskraft einer bestimmten oder bestimmbaren natürlichen Person zugeordnet werden können.

Die Weitergabe, Speicherung und Auswertung der Daten erfolgt nach gesetzlichen Bestimmungen ausschließlich ohne Namensnennung. Sie haben das Recht auf Auskunft (einschließlich unentgeltlicher Überlassung einer Kopie) über die erhobenen Daten, sowie das Recht, Berichtigung oder Löschung zu verlangen, soweit diese Daten noch nicht in die wissenschaftliche Auswertung eingegangen sind.

Die Aufbewahrung der erhobenen Daten erfolgt so lange, wie dies für den Nachweis der wissenschaftlichen Qualitätssicherung notwendig ist.

Für den Datenschutz in dieser Studie ist Frau Prof. Dr. med. Astrid Bertsche, Universitätsmedizin Rostock, Kinder- und Jugendklinik, Ernst-Heydemann-Str. 8, 18057 Rostock, zuständig.

Der Datenschutzbeauftragte der Universitätsmedizin Rostock ist erreichbar unter [email protected] (Universitätsmedizin Rostock, Doberaner Str. 142, 18057 Rostock, Tel.: 0381-494-5155).

Sie haben das Recht, beim Landesbeauftragten für Datenschutz und Informationsfreiheit Mecklenburg Vorpommern (Schloss Schwerin, Lennéstraße 1, 19053 Schwerin) Beschwerde einreichen.

Sponsor: Diese Studie wird nicht durch Dritte finanziell gefördert oder anderweitig finanziell unterstützt.

Bei Fragen sind wir jederzeit gerne für Sie unter [email protected] da.

Ihre Mitarbeiterinnen und Mitarbeiter des Projektes

Leiter des Forschungsvorhabens und Ansprechpartner:

Uni.-Prof. Dr. med. Astrid Bertsche, Kinder- und Jugendklinik der Universitätsmedizin Rostock, Ernst-Heydemann-Str. 8, 18057 Rostock

Weitere Ansprechpartner:

Ruth Melinda Müller ([email protected]), Kinder- und Jugendklinik der Universitätsmedizin Rostock, Ernst-Heydemann-Str. 8, 18057 Rostock

Birthe Herziger, Kinder- und Jugendklinik der Universitätsmedizin Rostock, Ernst-Heydemann-Str. 8, 18057 Rostock

Sarah Jeschke, Kinder- und Jugendklinik der Universitätsmedizin Rostock, Ernst-Heydemann-Str. 8, 18057 Rostock

Einwilligungserklärung

Einwilligungserklärung zur Teilnahme an dem Forschungsvorhaben

Untersuchung der intuitiven Anwendung von Notfallarzneimitteln durch medizinische Laien

Ich bestätige hiermit, dass ich durch Frau/Herrn                                                  (Name des Aufklärenden eintragen) mündlich über Wesen, Bedeutung, Risiken und Tragweite der beabsichtigten Beobachtungsstudie aufgeklärt wurde und für meine Entscheidung genügend Bedenkzeit hatte. Ich habe die Teilnehmenden-Information gelesen, ich fühle mich ausreichend informiert und habe verstanden, worum es geht.

Ich hatte ausreichend Gelegenheit, Fragen zu stellen, die alle für mich ausreichend beantwortet wurden. Ich hatte genügend Zeit mich zu entscheiden. Mir ist bewusst, dass ich jederzeit meine Zustimmung widerrufen kann.

Mir wurde mitgeteilt, dass für dieses Projekt keinerlei andere Arzneimittel als die vom Arzt ohnehin in der Routine verordneten zum Einsatz kommen. Alle Maßnahmen dienen ausschließlich der Qualitätssicherung der Behandlung mit Notfallarzneimitteln. Es erfolgt keine Intervention außerhalb der Routinebehandlung. Ein besonderes Risiko infolge der Teilnahme an diesem Forschungsvorhaben ist daher nicht gegeben.

Meine Einwilligung erfolgt ganz und gar freiwillig. Ich wurde darauf hingewiesen, dass ich meine Einwilligung jederzeit ohne Angabe von Gründen widerrufen kann, ohne dass mir dadurch irgendwelche Nachteile entstehen.

Informationen zum Datenschutz (personenbezogene Daten):

Bei wissenschaftlichen Projekten werden persönliche Daten über Sie erhoben. Die Weitergabe, Speicherung und Auswertung dieser projektbezogenen Daten erfolgt nach gesetzlichen Bestimmungen anonym.

Anonymisierung bedeutet, dass personenbezogener Daten derart verändert werden, dass die Einzelangaben über persönliche oder sachliche Verhältnisse nicht mehr oder nur mit einem unverhältnismäßig großen Aufwand an Zeit, Kosten und Arbeitskraft einer bestimmten oder bestimmbaren natürlichen Person zugeordnet werden können.

Mir wurde außerdem mitgeteilt, dass ich einerseits das Recht auf Auskunft (einschließlich unentgeltlicher Überlassung einer Kopie) über die mich betreffenden personenbezogenen Daten habe und andererseits, dass ich deren Berichtigung oder Löschung verlangen kann, soweit diese Daten noch nicht in die wissenschaftliche Auswertung, die keine personenbezogenen Daten enthält, eingegangen sind.

Für den Datenschutz in dieser Studie ist Frau Prof. Dr. med. Astrid Bertsche zuständig.

Der Datenschutzbeauftragte ist Herr Axel Peter (Kontaktdaten: Universitätsmedizin Rostock, Doberaner Str. 142, 18057 Rostock, Tel.: 0381-494-5155).

Ich habe das Recht, beim Landesbeauftragten für Datenschutz und Informationsfreiheit Mecklenburg-Vorpommern (Schloss Schwerin, Lennéstraße 1, 19053 Schwerin) Beschwerde einzureichen.

Bei Rücktritt von der Studie bin ich mit der Auswertung meines bisherigen Datenmaterials einverstanden. Andernfalls werde ich dies schriftlich oder mündlich beim Ansprechpartner für die Studie angeben.

Ich habe eine Kopie der Patienteninformation und dieser Einwilligungserklärung erhalten.

Unterschriften:

_____________________________                         _____________________________

(Ort, Datum)                                                                                 (Unterschrift Teilnehmer*innen [w/m/d])

_____________________________

(Unterschrift aufklärendes Personal)

Reviewer 2 Report

Comments and Suggestions for Authors

Study design (lines 99-103):  If I understand this correctly, subjects were not given the standard packages for these drugs/devices, and were not given any indication of how to use them (no diagrams, no written instructions). Instead, it seems that subjects were asked to infer proper use solely from an examination of the device. But why? What we want to know is whether parents would use these devices properly in the real world, a world in which packages would typically have diagrams and instructions. But this study is examining something very different: how parents would behave in a hypothetical world in which drugs/devices are packaged without instructions for use (or, at least, a world in which devices are always separated from their packaging). This makes the study less valuable. Why not combine this with a study of how well (and perhaps how quickly) parents can use the devices when standard packaging is present? (Lines 228-230 raise this issue, noting that parents are unlikely to seek guidance from healthcare professionals. True, but surely they'll look at the device packaging -- packaging that was not present for them in this study. The authors themselves make a similar point in lines 258-260.) This should be noted in the Limitations section of the manuscript.

Table 3:  The last column (number of parents making errors) is confusing. Where do the denominators of these fractions come from? Why are they not the same as the n for each device as shown in table 1? More explanation would help here.

This paper makes a new contribution to the literature on proper use of drug administration devices, but because study participants were not provided with the device packaging the significance of this contribution is somewhat diminished. This study answers the question, "If you were given this device and told nothing about it, could you use it correctly?" But was anyone asking that question? Aren't we more interested in this question: "When given a device in its usual packaging, could you use it correctly?"

Although this contribution is modest, it's still useful. It is worth publishing, but it's a close call.

Author Response

(The authors gave the same response as above.)

Reviewer 3 Report

Comments and Suggestions for Authors

The manuscript performed a detailed study on objective observation and subjective self-assessment of pediatric emergency 2 medication devices for parents. The perspectives of the study are thorough and considerate. Only a few things need to be addressed:

1.      Did the study consider parental anxiety level as a common reason for not conducting the procedures?

2.      In General Considerations, can authors conclude that as the complexity of the procedure or devices increases, the rate of parental intuition decreases?

3.      Can authors make a conclusive discussion why the different devices draw different reactions from the parents?

Comments on the Quality of English Language

Minor editing of English language required.

Author Response

(The authors gave the same response as above.)
